# Does the Measurement of Ejection Fraction Still Make Sense in the HFpEF Framework? What Recent Trials Suggest

**DOI:** 10.3390/jcm12020693

**Published:** 2023-01-15

**Authors:** Alberto Palazzuoli, Michele Correale, Massimo Iacoviello, Edoardo Gronda

**Affiliations:** 1Cardiovascular Diseases Unit Cardio Thoracic and Vascular Department, S. Maria alle Scotte Hospital University of Siena, 53100 Siena, Italy; 2Cardiology Unit, Policlinico Riuniti University Hospital, 71122 Foggia, Italy; 3Department of Medical and Surgical Sciences, University of Foggia, 71122 Foggia, Italy; 4Medicine and Medicine Sub-Specialties Department, Cardio Renal Program, UOC Nephrology, Dialysis and Adult Renal Transplant Program, IRCCS Ca’ Granda Foundation, Ospedale Maggiore Policlinico, 20122 Milano, Italy

**Keywords:** HFpEF, ejection fraction, cardiac function, management, phenotyping

## Abstract

Left ventricular ejection fraction (LVEF) is universally accepted as a cardiac systolic function index and it provides intuitive interpretation of cardiac performance. Over the last two decades, it has erroneously become the leading feature used by clinicians to characterize the left ventricular function in heart failure (HF). Notably, LVEF sets the basis for structural and functional HF phenotype classification in current guidelines. However, its diagnostic and prognostic role in patients with preserved or mildly reduced contractile function is less clear. This is related to several concerns due to intrinsic technical, methodological and hemodynamic limitations entailed in LVEF measurement that do not describe the chamber’s real contractile performance as expressed by pressure volume loop relationship. In patients with HF and preserved ejection fraction (HFpEF), it does not reflect the effective systolic function because it is prone to preload and afterload variability and it does not account for both longitudinal and torsional contraction. Moreover, a repetitive measurement could be assessed over time to better identify HF progression related to natural evolution of disease and to the treatment response. Current gaps may partially explain the causes of negative or neutral effects of traditional medical agents observed in HFpEF. Nevertheless, recent pooled analysis has evidenced the positive effects of new therapies across the LVEF range, suggesting a potential role irrespective of functional status. Additionally, a more detailed analysis of randomized trials suggests that patients with higher LVEF show a risk reduction strictly related to overall cardiovascular (CV) events; on the other hand, patients experiencing lower LVEF values have a decrease in HF-related events. The current paper reports the main limitations and shortcomings in LVEF assessment, with specific focus on patients affected by HFpEF, and it suggests alternative measurements better reflecting the real hemodynamic status. Future investigations may elucidate whether the development of non-invasive stroke volume and longitudinal function measurements could be extensively applied in clinical trials for better phenotyping and screening of HFpEF patients.

## 1. Introduction

Since the beginning of the last century, heart failure (HF) has been recognized as ‘a condition in which the heart fails to deliver its contents adequately’ [1]. In 1962, Folse and Braunwald showed that the ratio between left ventricular stroke volume (SV) and the end-diastolic volume (EDV) could provide significant information for a hemodynamic analysis of left ventricular function [2]. A few years later, Bartle proposed the term ‘ejected fraction’ to describe the ratio of SV to EDV [3].

Analysis of hemodynamic data gathered in different clinical contexts made clear that the HF syndrome was characterized by high cardiac filling pressures coupled with a reduced proportion of left ventricular stroke volume. The evidence supported the use of left ventricular ejection fraction (LVEF) as a suitable index to assess the ventricular function, although it mainly reflects the left ventricular volume that is dependent on the occurrent preload and the afterload [4]. Therefore, its changes do not reflect changes in the degree of myocyte shortening property and cannot measure myocardial contractility [5]. Initial randomized trials such as the Veterans Administration Cooperative Study (V-HeFT-I) [6] and the Cooperative North Scandinavian Enalapril Survival Study (CONSENSUS) had as inclusion criteria, either radiologic parameters such as cardiac size, left ventricular diameter in diastole >2.7 cm/m^2^ and LVEF <45% on echocardiography with sign and symptoms typical for heart failure (HF) [7,8]. Subsequently, the SOLVD trials first introduced LVEF as inclusion criteria [9]. Current analysis revealed that low LVEF values were associated with increased events without distinguishing among HF aetiologies, ischemic or not ischemic, substrate and arrhythmogenic risk burden. Since LVEF emerged as one relatively strong predictor, it was successively adopted in all HFrEF trials and also extended in patients with less advanced systolic dysfunction or isolated diastolic dysfunction. Interestingly, the lowest LVEF value was coupled with the largest benefit provided by beta blocker therapy, and subjects with the lowest systolic blood pressure quartile (≤95 mm Hg) experienced the largest proportion of benefit, together with significant increase in systolic blood pressure [10]. The rise of systolic blood pressure was the index of contractile restoration linked to biological effect of the attempted agents such as renin-angiotensin-aldosterone inhibitors (RAASi) and betablockers. Despite the LVEV is considered an index of cardiac contractility, and it became the leading metric feature used by clinicians for HFclassification, it is poorly related with hemodynamic profile and effective systolic function

## 2. Critical Analysis of HFpEF Trials Based on LVEF Threshold 

Multi-center trials have demonstrated consistent differences in patient screening and inclusion criteria.

Despite the limitations of LVEF, since the 1990s the occasional occurrence of HF symptoms plus the detection of LVEF ≥ 45% has been arbitrarily considered by international guidelines as the diagnostic criterion to qualify the diastolic left ventricular impairment as the prominent if not the exclusive cause of HF development [11,12].

Although the scientific statements were wrapped in confusing concepts, the first cardiovascular outcome study, titled “Effects of candesartan in patients with chronic heart failure and preserved left-ventricular ejection fraction: the CHARM-Preserved Trial”, was performed in order to prospectively investigate an angiotensin receptor blocker (ARB) in patients with diastolic dysfunction [13] (Table 1).

Of note, the study adopted the definition of “preserved ejection fraction” by using a cut-off for LVEF ≥ 40%. This could have led to an increase in the fraction of subjects with mild systolic dysfunction among the 3023 enrolled patients [14]. 

The study results did not clarify the therapy issue for treatment of HFpEF, but fueled the rationale to perform a study with another ARB molecule, irbesartan. That study was based on a population with a clinical profile close to that of the longitudinal observational Cardiovascular Health Study (CHS) [15]. Patients enrolled had normal left ventricular systolic function and no heart failure symptoms. Their rate of death per 1000 patient-years of follow-up was 25; conversely, it rose to 87 for those developing HFpEF.

These data influenced the profile of patients screened for entering into I-PRESERVE study, which was designed to enroll patients similar to a real word population, focusing on LVEF > 45%, NYHA II-IV, HF hospitalization <6 months and positive ECG or echocardiographic signs of left ventricular hypertrophy [16]. These restrictive criteria led to the enrollment of a study population prominently based on female gender 60%, aged 72 years and with high hypertension prevalence. The overall number of deaths was 881 (relatively high 5.23% rate per year) and cardiovascular (CV) mortality accounted for 532 deaths (60%) [17]. Surprisingly, the overall hospitalizations were almost three times higher than HF hospitalization (respectively, 19.9 vs. 7.42).

The positive aspect achieved by CHARM and I-PRESERVE was the definition of HFpEF criteria addressed to better defining both patient characteristics and more appropriate management.

In a short time, the “Spironolactone for Heart Failure with Preserved Ejection Fraction” (TOPCAT) study followed. That study investigated a patient population aged 50 years or older, with LVEF > 45%, history of hospitalization within the previous 12 months and brain natriuretic peptide [BNP] level ≥ 100 pg/mL, or an N-terminal pro-BNP [NT-proBNP] level ≥ 360 pg/mL [18]. Despite the strict patient selection (female gender 50%, average age 72 years, LVEF 58%), spironolactone failed to reduce the composite outcome of CV disease, aborted cardiac arrest, or hospitalization, but the HF hospitalization rate was significantly reduced in the spironolactone group in comparison to the placebo group (HR 0.83; 95% CI, 0.69 to 0.99, *p* = 0.04).

Subsequently, a multicenter, randomized, double-blind, parallel group, active-controlled study was designed to evaluate the efficacy and safety of LCZ696 (sacubitril-valsartan) angiotensin renin neprilysin inhibitors (ARNI) compared to Valsartan in HFpEF (PARAGON-HF) [19]. The study entry criteria were meticulous in qualifying the cardiac condition for patient screening: age ≥ 50 years, with a LVEF ≥ 45% within the 6 months before randomization and without any prior echocardiographic measurement of LVEF < 40%, NYHA II - IV required for at least 30 days before screening, and they had to have NT-proBNP > 200 pg/mL if the patient had been hospitalized for HF within the past 9 months, or >300 pg/mL without a recent hospitalization The NT-proBNP requirement was tripled if patients had atrial fibrillation (AF). In addition, patients had evidence of structural heart disease, including either LV hypertrophy or LA enlargement [20].

Notably, the high cut-off of NT-proBNP at enrollment was based on evidence that in the I-PRESERVE trial the plasma high concentration of the peptide was linked to the worse outcome. On the other hand, the study sample size was calculated through simulations for the proportional rates model, and the candesartan group of the CHARM-Preserved study, involving patients with EF ≥ 45%, provided the rates for statistical assumption. That decision seemed to contradict the adopted rigorous patient selection criteria, since in the CHARM-Preserved trial the LVEF value was adopted as an entry criterion at the time of enrollment without evaluating previous measurement. Therefore, the medical therapy did not include beta blockers and minaral corticoid receptor antagonists (MRA); conversely, digitalis was prescribed as a first line therapy. 

The study randomized 4822 patients (mean age 73 years, 52% females, average LVEF 58%) to receive sacubitril valsartan or valsartan, and narrowly missed the combined primary endpoint to decrease HF hospitalization and CV death (O.R. 0.87; 95%:CI, 0.75 to 1.01; *p* = 0.06) [21]. 

The study’s subgroups analysis for the composite primary endpoint amazingly addressed female gender (RR 0.73: CI 0.58–0.90) and patients with LVEF ≤ 57% (RR 0.78: CI 0.64–0.95). The benefit size in both subgroups was comparable to that achieved by the HFrEF population investigated in the PARADIGM HF study (RR 0.79: CI 0.71–0.89) [20], highlighting that a normal LVEF does not address the absence of systolic dysfunction and that the female gender reacts differently to medical therapy, probably due to heart reverse remodeling capability [21,22].

One question that remained unanswered was what would be the response of HF patients presenting LVEF higher than 57% when HFrEF effective therapies were adopted. In the case of the absence of systolic dysfunction, the literature addresses the issue that beta-blocker treatment does not provide significant benefit [23,24].

A striking benefit was displayed by the sodium-glucose cotransporter inhibitors (SGLT2i) in HFrEF diabetic and non-diabetic patients with high CV risk [25]. There was a significant decrease in HF hospitalization, fueling the decision to perform the EMPagliflozin outcomE tRial in patients with chrOnic heaRt failure, the EMPEROR-Preserved study. 

The trial design aimed to demonstrate the superiority of empagliflozin 10 mg versus placebo in patients with symptomatic, chronic HFpEF, addressed by the LVEF > 40%, and under stable treatment of HF symptoms. In the protocol, the patient randomization had to be stratified by the following factors: status of diabetes at screening, eGFR (CKD-EPI) at screening <60 mL/min/1.73 m^2^ or ≥60 mL/min/1.73 m^2^, and LVEF cut-off subgroups for the efficacy end points <50%, between 50–60%, and >60% [26].

The decision to make the change was probably driven by the publication of the combined data from PARADIGM-HF (LVEF eligibility ≤ 40%; n = 8399) and PARAGON-HF (LVEF eligibility ≥ 45%; n = 4796) in a prespecified pooled analysis at the end of 2020 [27]. In the manuscript, the ARNI studies population was stratified with similar cut-off subgroups. The investigation results reinforced the evidence that the ARNI benefit was in a higher LVEF range for women compared with men. 

The EMPEROR Preserved study was performed on 5988 patients, aged 72 years, women were 45%, the median LVEF was 54%, nearly half of the patients had diabetes and half had eGFR reduction below 60 mL/min/1.73 m^2^. The main study end point based on CV disease and HF hospitalization was successfully achieved (HR 0.73; 95% CI, 0.6–0.88; *p* < 0.001), and this was the first time a drug therapy reached the goal in an HFpEF study [28].

According to the prespecified baseline LVEF subgroups, the best therapy benefit was achieved in the cluster with LVEF < 50% (HR 0.71; 95% CI, 0.57–0.88), while the benefit was attenuated in patients with LVEF 50 to <60% (HR 0.80; 95%; CI, 0.64–0.99) and was not evident in the subgroup with LVEF ≥ 60% (HR 0.87; CI, 0.69–1.10). In a further pooled analysis performed on both the EMPEROR-Reduced and EMPEROR-Preserved trials, patients were grouped based on LVEF: <25% (n = 999), 25–34% (n = 2230), 35–44% (n = 1272), 45–54% (n = 2260), 55–64% (n = 2092), and ≥65% (n = 865). The analysis showed that the magnitude of the effect of empagliflozin on HF outcomes was similar in patients with ejection fractions <25% to <65%, but it was attenuated in patients with upper values [29].

A potential limit of EMPEROR-Preserved was due to the relative minority of female gender, and the low NYHA class in the investigated population. Additionally, the lack of benefits of empagliflozin for overall mortality was observable (HR 1; 95% CI, 0.87–1.15). These items reinforced the post hoc PARAGON-HF study message that in elderly subjects with HF symptoms, CV death is more frequent among those in the lower LVEF range, whereas other conditions increasingly affect the survival of subjects with higher LVEF values [30]. 

Finally, the DELIVER trial evaluated 6263 patients with LVEF > 45%, NT-proBNP levels > 225 pg/mL, or 375 pg/mL in patients with atrial fibrillation, with GFR > 25 mL/min/m^2^, requiring diuretic treatment or recent hospitalization or urgent visit requiring IV therapy or evidence of structural abnormalities at echocardiography [31]. Demographic characteristics showed 44% were female, the mean LVEF was 54%, the mean age 71 years, the mean NT-proBNP was 1011 pg/mL and 18% had improved LVEF. The established primary end point was combination of CV death and worsening HF [32]. During the follow up (mean of 2.3 years), dapaglifozin demonstrated an 18% relative risk reduction (HR 0.82; CI: 0.73–0.92), along with a non-statistical significance in overall CV mortality reduction (HR 0.88; CI 0.74–1.05). 

Subgroup analysis revealed that the benefit was consistent beyond the presence of diabetes and impaired renal function [32]. Therefore, the benefit was maintained irrespective of NYHA class, although more advanced stages demonstrated a greater improvement in quality of life [33]. 

Differently from previous trials, the DELIVER study design included data showing that risk reduction was maintained across all LVEF range; however, authors did not report if the benefit was related to reduction of total CV events, HF events, or both, in subjects with LVEF > 60%. 

Altogether, the findings on SGLT2i raise some burning questions: Despite identical results on major events for both empaglifozin and dapaglifozin, what is the LVEF threshold providing patient benefit? In the upper LVEF values, was risk reduction related to a decrease in HF hospitalization or total mortality? And, are the positive drug effects consistent across all LVEF cut-offs irrespective of renal function? 

## 3. The Misleading Significance of Ejection Fraction

Although LVEF is the main feature for HF categorization, it has relevant mechanistic, methodological and intrinsic limitations.

LVEF is the most common practical measurement to assess cardiac pump function, and for this reason it is recognized as the hallmark for systolic function classification. Notably, LVEF offers some advantages related to its ease of use, its wide application in clinical practice and study research, the short scan time, and its feasibility [34]. Therefore, EF can be calculated easily and it can be assessed visually, even without specific expertise [35]. An additional advantage is the restricted variation in normal physiological conditions, and its independence of body weight, size, and race, although it is slightly higher in women compared to men. 

Moreover, LVEF provides the basis for structural and functional phenotype classification, and it is particularly useful in patients with reduced systolic function [36,37]. However, its role in patients with preserved or mildly reduced contractile function is less clear. This is related to several gaps due to intrinsic mechanistic, methodological and hemodynamic limitations that do not appropriately describe the real contractile performance and pressure volume loop relationship [38]. For these reasons, the role of EF as the main determinant to classify patients with HF has been recently questioned (Table 2).

Since LVEF is a load-dependent measure, it is prone to changes in preload and afterload. Of note, the elevation or reduction in both systemic blood pressure or vascular stiffness can result in relevant LVEF changes [39]. Conversely, a reduction in preload, causing a decrease in the LV filling blood flow, leads to LV wall strain forces reduction together with cardiac output decrease. In the presence of a valve defect, LVEF may be over- or underestimated; in the case of significant mitral regurgitation, this is due to the reduced workload during cardiac contraction. Alternatively, in the setting of aortic stenosis, an increase of afterload occurs along with a delay in outflow time peak and consequent LVEF reduction [38,40]. Systemic arterial and pulmonary venous pressure are two main determinants of afterload and preload, respectively, but other factors such as intrinsic myocyte forces, distension capacity, chronotropic incompetence, ventriculo-arterial coupling and pressure-volume curve adaptation during exercise, are all potential confounders for LVEF estimation [41]. Additionally, LVEF is prone to heart rate variability; significant reduction implies a prolonged diastolic time, a better filling flow, and consequent increase in cardiac output in physiologic condition. Conversely, increased heart rate reduces filling time, diastolic volume is less expanded, and LVEF could be underestimated. Similarly, sympathetic activity or vagal stimulation, together with other systemic conditions such as anaemia, thyroid dysfunction, endocrine and metabolic alterations, are all features that could potentially influence LVEF assessment [42,43].

Beyond these methodological observations, systolic function was initially calculated measuring stroke volume divided by end diastolic volume. This approach takes into consideration both a parameter of ejected flow and LV remodelling. Since LVEF calculated with the traditional method is simply the ratio between LV end diastolic volume (EDV) and LV end systolic volume (ESV), it significantly decreases when EDV is in the normal range or in LV concentric hypertrophy. Conversely, in LV eccentric remodelling, in which EDV enlargement acts as compensatory mechanism, LVEF is maintained into the normal range in the presence of identical ESV dilatation. Thus, impaired LVEF does not necessarily imply a reduced stroke volume, although LVEF impairment is often associated with an increase in LV filling pressure [43,44].

These assumptions prove that LVEF is not necessarily related to the cardiac output, and it has been erroneously considered as an indicator for LV remodelling. Indeed, an enlargement of diastolic dimension works as a compensatory factor in order to maintain an adequate stroke volume even during the occurrence of increased systolic volume [45,46]. Notably, LVEF is inversely related to ESV, but poorly related to stroke volume; thus, it is a mirror of systolic dysfunction in eccentric remodelling, but in the absence of relevant EDV dilatation, such as HFmEF or HFpEF phenotypes, it does not reflect effective contractile status. On the other hand, stroke volume remains relatively constant over a wide range of LV remodelling, and it is the true parameter of effective cardiac pump function [4,47]. 

Despite LVEF being considered a measurement of systolic function, it really measures only radial contraction without measuring both longitudinal and circumferential contractility, representing two important features of the whole systolic performance. Accordingly, several studies including patients with preserved LVEF have shown significant longitudinal global function impairment, despite preserved ejection fraction [48,49]. The radial contraction measured by LVEF is an index of sub-endocardial and mid-wall fibers contraction, but the more external layers’ contribution to LV chamber contraction are not detected [48]. For these reasons, LVEF cannot be considered a precise measure of global systolic function, and it may be matched with different cardiac remodelling, loading conditions, filling pressure, systemic resistances and sarcomeric forces [49,50]. 

The most common modality of LVEF measurement is provided by echocardiography, adopting the biplane Simpson calculation. It is recognized as the most reliable method with a good relationship to the angiography measurement [38,51]. This formula is based on the assumption of precise LV elliptic shape and regular conformation and it includes a constant value; however, in the case of spherical geometry or pyramidal conformation, the final results cannot precisely reflect the effective LV volumes. Many studies have demonstrated an overestimation of echocardiography compared to cardiac remodelling in terms of LV volume, as the common biplane method has intrinsic limits related to standard shape assumption which lead to a more imprecise measurement [50]. Unfortunately, the inter-observer variability even in accredited echocardiography laboratories ranges from 5 to 8%, with broader limits for less experienced physicians. Thus, the current ESC cut-off distinguishing HFrEF for patients with LVEF < 40%, mid-range for patients with LVEF between 40 and 49%, and HFpEF for those with LVEF > 50%, makes this classification hard to distinguish [52]. A relevant variability exists among different imaging modalities commonly employed for LVEF estimation. Indeed, correlations among the different techniques show modest accuracy and a wide cut-off range. On the other hand, the reproducibility is high for cardiac magnetic resonance, which applies a three-dimensional reconstruction and exact LV shape reproduction [53,54]. Unfortunately, with the most widely available techniques, such as echocardiography, SPECT or cardiac CT, the LVEF estimation is less accurate and precise. Finally, echocardiography has some intrinsic limitations related to endocardial border visualization, papillary muscle cutting, and exact atrio ventricular plane definition due to mitral anulus movement during cardiac cycle [55,56,57] (Figure 1).

## 4. The Dynamic Changes of Left Ventricular Ejection Fraction across Heart Failure Journey 

Since HF is a transitional syndrome, LVEF may change across a patient’s history according to disease destabilization and hemodynamic fluctuations.

ESC guidelines, and more recently ACC recommendations, have confirmed a new HF categorization theoretically capable of identifying patients with mildly reduced ejection fraction in which the cut-off ranges from 40 to 49% [52,55]. Indeed, the current classification is an attempt to identify a specific biological and pathophysiological cluster in subjects with clinical manifestations typical of HF, increased natriuretic peptides and moderate cardiac systolic dysfunction. Perhaps HFmEF is a mixed model, containing patients with mixed phenotypic and clinical characteristics typical of both reduced and preserved LVEF [58]. Despite this new categorization, HFmEF remains poorly represented in most clinical trials and relatively characterized in terms of etiology, history and underlying pathophysiological mechanisms. Current discrepancies arise from the indeterminate profile in this subgroup including both HFpEF and HFrEF features, and it probably subtends different trajectories and prognoses [59]. Original data come from single center studies with unrepresentative sample size and limited diagnostic criteria based only on LVEF threshold. Clinical characteristics, CV risk profile, extracardiac comorbidities and echocardiographic features are often neglected. The CHARM-Preserved study which included patients with LVEF > 40% showed that patients with mildly reduced LVEF were much more often females with intermediate mean age values and hypertension prevalence between HFrEF and HFpEF [23]. Therefore, HFmEF had a similar prevalence of coronary artery disease (CAD) and atrial fibrillation (AF) compared to HFrEF, whereas creatinine values and NYHA class distribution were intermediate between HFrEF and HFpEF. Despite different clinical characteristics, the HFmEF group revealed a reduced trend of HF hospitalization and death for CV causes compared to HFrEF.

The retrospective analysis of the DIG trial demonstrated that HFmEF resembled patients with HFrEF in terms of mean age, sex and ischemic aetiology [60]. In the TOPCAT trial, which included patients with mean LVEF above 45%, the mean age, female prevalence and hypertension were higher in those with mildly reduced LVEF, whereas other comorbidities such as chronic kidney disease (CKD), CAD, AF and diabetes were similar between groups [61]. Interestingly, an Asian registry revealed different AF prevalence with linear increase according to LVEF: 29% in reduced, 40% in mildly reduced, and 45% in preserved LVEF [62]. The ESC observational registry reveals that HFmEF patients have characteristics similar to HFpEF in terms of age, female prevalence and hypertension, although CAD prevalence resembled the prevalence occurring in HFrEF. Morality rate at one year significantly differed between HFpEF and HFmEF (6.3 vs. 7.6%, respectively) [63]. A study focusing on extracardiac and demographic characteristics matched for LVEF demonstrated an increased burden of extracardiac diseases, with higher prevalence of lung diseases and diabetes, in patients with high LVEF [64]. In a recent Swedish registry analysis comparing three common comorbidities such as AF diabetes and CKD, HFmEF revealed an intermediate prevalence of CKD and AF, whereas diabetes was similarly expressed in all HF groups [65]. Finally, the combined analysis of PARADIGM and PARAGON confirmed an intermediate range regarding age, female sex, body mass, natriuretic peptides and hypertension, whereas history of myocardial infarction resembled HFrEF [27]. All together, these findings suggest that HFmEF is an intermediate or transitional status between HFpEF and HFrEF. Indeed, this group may be a combination of patients with initially preserved EF who are heading towards a more relevant systolic dysfunction; on the other hand, patients demonstrating a good response to medical or electrical therapies may experience a significant improvement in LVEF associated with cardiac reverse remodeling. The current mixture can reflect different baseline characteristics in terms of risk factors, CAD prevalence and extracardiac comorbidities in different trials. The opposite underlying pathophysiological process (from concentric to eccentric remodeling vs. eccentric to reverse remodeling) could affect response to treatment and long-term outcome [66]. Notably, a meta-analysis reports different prognostic trajectories with neurohormonal antagonism, confirming the heterogeneity of this phenotype [67]. Therefore, a mildly reduced LVEF does not necessarily represent a unique phenotype and it does not exclusively entail the maladaptive mechanisms seen in poor responder HFpEF patients [68]. LVEF may change even with physiologic condition. Indeed, some endurance athletes reveal an LVEF cut-off below the normal limit at rest that then becomes super normal during exercise with preload and afterload changes; peripheral vasodilation related to muscle requests and increased venous return yields to a contractile function recover when body metabolism and energetic requirement enhance [69]. Beyond these features, the HFpEF classification as LVEF > 50 % appears misleading and simplistic because it does not take into account several factors such body conformation, risk factors cluster, associated diseases, and vascular and cardiac remodeling [70]. Accordingly, a recent cluster analysis from the SwedeHF registry attempts to better phenotype the HFpEF population, demonstrating at least five main distinct clusters [71].

## 5. The Favourable Data Coming from HFpEF Trials beyond Ejection Fraction

Recent interventional trials and metanalysis have demonstrated a trend towards outcome improvement, and current findings are mainly related to total cardiovascular event reduction.

HF is a syndrome with continuous disease progression and evolution related to functional and structural cardiac and extra-cardiac changes. Current knowledge leads to a wide phenotype spectrum with frequent overlapping [72]. A novel HF classification could be developed from new technologies and scientific discoveries that will allow a deeper understanding of HF beyond LVEF [73]. A multi-modality model including cardiac function assessment, vascular adaptations, detailed environmental and lifestyle information, and laboratory and biomolecular fingerprints coming from big data analysis may clarify the distinct phenotype and address a specific therapeutic response.

HFpEF syndrome, characterized by an interplay of genetic predisposition, lifestyle factors and high burden of associated comorbidities, is proof that other parameters (congestion, kidney function, metabolic and vascular stiffness) matter more than LVEF [74].

SGLT2i treatment has been demonstrated to improve CV outcomes in patients with HF over a wide range of LVEF, regardless of diabetic status, and has a strong reno-protective effect [75].

Results from the DELIVER [30] and EMPEROR-Preserved trials [26] with SGLT2i in patients with HFpEF (LVEF > 40%) encourage the use of SGLT2i for the pharmacological approach to HFpEF [76]. However, a pooled analysis performed on both the EMPEROR-Reduced and EMPEROR-Preserved trials demonstrated that empagliflozin reduced the risk of HF hospitalization by ≈30% in all LVEF subgroups, with an attenuated effect in patients with an LVEF ≥ 65% [29] (Figure 2).

Regarding the effects of therapy combinations across the LVEF spectrum, by using pooled data from PARADIGM-HF [21] and PARAGON-HF [19] and combining data with studies investigating MRAs and the EMPEROR program, Vaduganathan et al. [77] showed that the switch from RAASi to ARNI, combined with the addition of MRA, and the SGLT2i, were associated with a reduction in the composite outcome for subgroups with LVEF ranging from 45% to 54% and LVEF 55% to 64%, but not in those with higher values.

Data derived from DELIVER [30] and EMPEROR-Preserved [26] demonstrated that the improved outcome is related to improved renal function in patients treated with gliflozins. According to what was stated by Lam and Solomon [78], the mechanism of action for SGLT2i in HF is different from that of all the other HF therapies that are focused on reverse left ventricular remodeling, blocking the neurohormonal activation, or enhancing the natriuretic peptide system. Even if these agents have been shown to have favorable kidney effects, this trend mainly depends on systemic perfusion improvement and direct action in reducing left ventricular size and cardiac performance [79]. Instead, reverse left ventricular remodeling with SGLT2 inhibitors is actually not very significant despite evidence of their good outcome. That the outcome improved was recently confirmed in a prespecified meta-analysis comparing DELIVER and EMPEROR-Preserved results with trials evaluating SGLT2i effects in HFrEF; the study demonstrated that reduction of CV events and HF hospitalization is clear irrespective of LVEF, and current findings may change the paradigm of HF categorization by this parameter [80]. Additionally, the beneficial evidence of these agents is connected with several features such as BMI, renal function, NYHA class and natriuretic peptide level, confirmed a class effect that makes irrelevant the distinction between HFrEF and HFpEF.

The SGLT2i are reno-protective and vascular-protective, and kidney effects seem to predominate over all the other mechanisms of action.

The benefit of SGLT2i goes beyond glycemic control, since inhibition of renal glucose reabsorption affects blood pressure and improves the hemodynamic profile and the tubule glomerular function [81]. SGLT2i may address kidney function and improve hemodynamic homeostasis, opposing the congestive state regardless of LVEF.

The improvement in renal function may justify the reduction in adverse outcomes in patients with or without HF, supporting the results of clinical trials. In fact, the efficacy of SGLT2i was mainly linked to the reduction in renal damage, as highlighted by the significant reduction in global mortality in the studies on diabetic and non-diabetic populations with advanced chronic kidney disease (CKD) [82]. A renal benefit from SGLT2i use in patients with HFrEF was shown. SGLT2i therapy has been found to substantially relieve cardiorenal morbidity in patients with CKD or HFrEF, regardless of the presence of T2DM and the severity of CKD or HF [83]. Their cardiorenal benefits are present across a range of eGFRs (within CKD1–3 groups) and the presence or absence of ischemic heart disease, HF or T2D.

Unlike traditional loop diuretics such as furosemide leading to a neurohormonal activation, natriuresis derived from Empagliflozin is not associated with neurohormonal activation or potassium loss or renal impaired function, and is usually associated with an improvement in uric acid levels. This favorable diuretic profile provides an important advantage in the management of patients with HFrEF and HFpEF and could partly explain the better long-term HF outcomes [79].

The benefits of SGLT2i may also be due to favorable fatty acid, glucose uptake and metabolic effects on LV. The direct effects of SGLT2i on the endothelial and vascular function may contribute to hemodynamic effects [84]. There are also favorable metabolic mechanisms, including increased insulin sensitivity with glucose uptake in muscle cells, decreased gluconeogenesis and increased ketogenesis [85]. SGLT2i use improves cardiac energetic metabolism and slows myocardial oxidative stress injury in patients with HF, from which there is a mild state of insulin resistance with free fatty acids (FFAs) as principal energy source. The increased ketogenesis improves cardiac function [86].

## 6. Conclusions

LVEF remains a universal parameter employed for macroscopic HF categorization but it contains several methodological and hemodynamic weaknesses. Recent trials have demonstrated that the benefit of treatment is consistent regardless of the LVEF value.

The dynamic nature and ambiguity of HF requires repetitive assessment, looking at disease temporal changes depending on the natural history of diseases and the response to therapy. Additionally, traditional and new drugs may have different positive effects on HFpEF patients; in those with borderline LVEF, the risk reduction is related to an effective decrease in HF events, whereas in patients with higher LVEF values, the main benefit is due to a decrease in CV events. Because of these findings, future research may shift to different and more reliable cardiac functional parameters. Therefore, cardiac dysfunction should be assessed within a larger context, looking at the underlying causes of HF, its temporal trend and cardio-renal axis status.

## Figures and Tables

**Figure 1 jcm-12-00693-f001:**
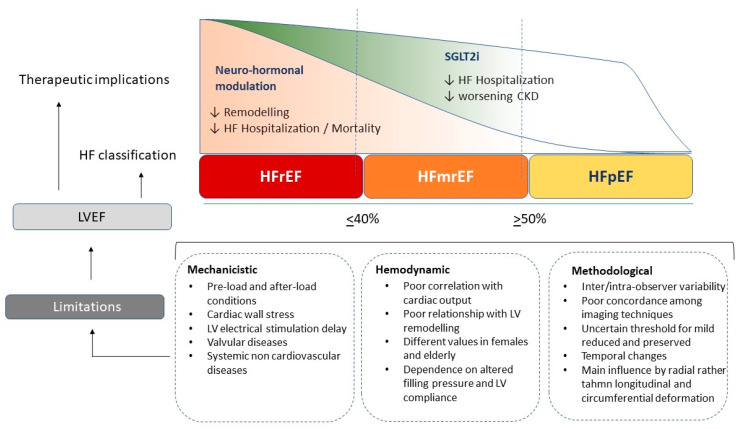
Different effects of current heart failure treatments on renin-angiotensin-aldosterone axis (RAAS), sympathetic activation and reno-protective effects across the range of left ventricular ejection fraction; ARNI angiotensin renin neprilysin inhibitors, SGLT2i sodium-glucose transporters-2 inhibitors, MRA mineral-corticoid receptor antagonist.

**Figure 2 jcm-12-00693-f002:**
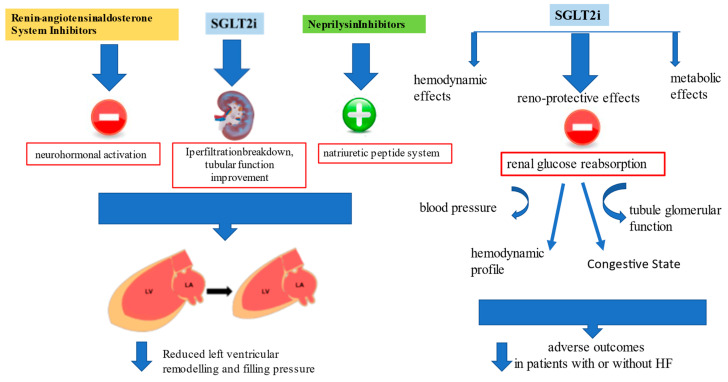
Renin-angiotensin-aldosterone system inhibitors (RAASi) and neprylisin antagonist actions on cardiac remodeling and progressive dysfunction, and additional metabolic, vascular and renal protective effects of sodium-glucose transporters-2 inhibitors (SGLT2i) beyond functional cardiac status improved outcome in both HFrEF and HFpEF.

**Table 1 jcm-12-00693-t001:** The main studies evaluating the efficacy of pharmacologic approaches in patients with heart failure and mildly reduced or preserved ejection fraction; different inclusion criteria and patient characteristics may affect primary outcomes.

Study	Drug Tested	LVEF	Other Main Enrollment Criteria	Number of Patients	Follow-Up	Results
PEP-CHF	Perindopril vs. placebo	>40%	- >70 years - cardiovascular hospitalization within the previous 6 months - absence of hemodynamically significant valve disease - systolic arterial pressure > 100 mmHg - potassium < 5.4 mmol/L	852	Minimum 1 year	- No significant change in primary end point (all-cause mortality and unplanned heart failure-related hospitalization): HR 0.919: 95% CI 0.700–1.208; *p* = 0.545
I-Preserve	Irbesartan vs. placebo	>40%	- NYHA class II-IV - pulmonary congestion on radiography - left ventricular hypertrophy or left atrial enlargement on echocardiography - left ventricular hypertrophy or left bundle-branch block on electrocardiography - SAP ≥ 100 mm Hg - creatinine < 2.5 mg/dL	4128	Mean 49.5 months	- No significant reduction of the primary composite end point (death from any cause or hospitalization for a cardiovascular cause, i.e., heart failure, myocardial infarction, unstable angina, arrhythmia, or stroke): HR: 0.95; 95% CI: 0.86–1.05; *p* = 0.35 - No significant reduction of HF hospitalizations: HR 0.95; 95% CI: 0.85–1.08; *p* = 0.44
CHARM-Preserved	Candesartan vs. placebo	>40%	- NYHA class - history of hospital admission for a cardiac reason - all treatments other than angiotensin-receptor blockers	3023	Median 36.6 months	- No significant change in primary end point (all-cause mortality and unplanned heart failure-related hospitalization): HR 0·89; 95% CI 0.77–1.03; *p* = 0.118 - Reduction of hospitalization due to HF (*p* = 0.017)
TOPCAT	Spironolactone vs. placebo	≥45%	- controlled systolic blood pressure - serum potassium < 5.0 mmol/L - history of hospitalization within the previous 12 months - elevated natriuretic peptides (BNP/NTproBNP) - GFR > 30 mL/kg × 1.73 m^2^ or creatinine <2.5 mg/dL	3445	Median 3.3 years	- No significant change in the primary end point (death from cardiovascular causes, aborted cardiac arrest, or hospitalization for the management of heart failure): HR 0.89; 95% CI: 0.77–1.04; *p* = 0.14 - Significant reduction of HF hospitalizations: HR: 0.83; 95% CI: 0.69–0.99, *p* = 0.04
PARAGON-HF	Sacubitril/Valsartan vs.Valsartan	≥45%	- NYHA class II-IV - high NT-proBNP - GFR > 30 mL/kg × 1.73 m^2^	4822		- No significant change in primary end point (all-cause mortality and unplanned heart failure-related hospitalization): HR: 0.87; 95% CI: 0.75–1.01; *p* = 0.06
EMPEROR-preserved	Empaglifozin vs. placebo	>40%	- NYHA class II-IV - high NT-proBNP - left atrial enlargement or increased LV hypertrophy on echocardiography - GFR > 20 mL/kg × 1.73 m^2^	5988	Median 26.2 months	- Significant reduction of the primary end point (cardiovascular death or hospitalization for heart failure): HR: 0.79; 95% CI: 0.69–0.90; *p* < 0.001 - Significant reduction of total number of hospitalizations: HR: 0.73; 95% CI: 0.61–0.88; *p* < 0.001
DELIVER	Dapaglifozin vs. placebo	>40%	- NYHA class II-IV - left atrial enlargement or increased LV thickness - high NT-proBNP - GFR ≥ 25 mL/kg × 1.73 m^2^	6263	Median 2.3 years	- Significant reduction of the primary end point (cardiovascular death or unplanned hospitalization for heart failure or urgent visit for heart failure): HR: 0.82; 95% CI: 0.73–0.92; *p* < 0.001 - Significant reduction of unplanned hospitalization for heart failure or urgent visit for heart failure: HR: 0.79; 95% CI: 0.69–0.91

**Table 2 jcm-12-00693-t002:** Advantages and limitations of using LVEF: high feasibility and wide application are balanced by poor relationship to effective hemodynamic status and high variability.

LVEF in Heart Failure
Advantages
Wide application in clinical practice and study research.
High feasibility.
Restricted variation in normal physiological conditions.
Independent of body weight, size and race.
Relationship with the beneficial effects of neurohormonal modulation when reduced.
Disadvantages
Not a measure of cardiac contractility (expression of radial contraction rather than longitudinal and circumferential).
High variability among the available techniques for estimation.
High inter-observer variability in echocardiographic assessment.
Limited sensitivity in detecting systolic dysfunction when compared with 2-D strain.
Slight gender-related differences (higher in women compared to men).
Dependence on loading conditions and valvular function.
Influenced by heart rate.
Confusing role when mildly reduced or preserved.
Dynamic behavior during natural history of heart failure.

## Data Availability

Not applicable.

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
