# Peer review of "Does the Measurement of Ejection Fraction Still Make Sense in the HFpEF Framework? What Recent Trials Suggest"

_jcm, 2023, doi:10.3390/jcm12020693_

Round 1
Reviewer 1 Report
Overall, the paper is very well written. May I ask the authors to consider the following?
The title would benefit from rewording
A brief discussion about the low(er) EF in endurance athletes would complete the picture
Neither in the abstract nor the introduction does the purpose of this review become clear
The paper might be easier to read when beginning each chapter with the key message and then substantiating this with the detailed literature description.
There are some grammar errors and typos, i.e., lines 240, 252, 253, 339, 440…..
Starting the abstract with a preposition is unusual
Please be correct when using higher vs increased or lower vs decreased
Author Response
Overall, the paper is very well written. May I ask the authors to consider the following?
Many thanks for your appreciation, we changed the new version accordingly with your suggestions.
The title would benefit from rewording
Title is re-wrriiten in “ Does the measurement of EF still make sense into HFpEF Framework? What recent trials suggest” I hope this heading is more appropriate
A brief discussion about the low(er) EF in endurance athletes would complete the picture. This is not the topic of our manuscript however we inserted a brief sentence according to Reviewer comment within the section “Dynamic changes of Ejection fraction across Heart failure journey
“LVEF may changes even during physiologic condition, indeed some endurance athletes reveal LVEF cutoff below normal limit at rest that becomes super normal during exercise with preload and afterload changes: peripheral vasodilation related to muscle requests and increased venous return yields to a contractile function recover when body metabolism and energetic requirement enhance.” new ref 69 was added
Neither in the abstract nor the introduction does the purpose of this review become clear. We added the aim of manuscript into the abstract
The paper might be easier to read when beginning each chapter with the key message and then substantiating this with the detailed literature description. We inserted a key message at the beginning of each paragraph
There are some grammar errors and typos, i.e., lines 240, 252, 253, 339, 440….Sorry for grammar and typos mistakes have been corrected .
Starting the abstract with a preposition is unusual Deleted Despite
Please be correct when using higher vs increased or lower vs decreased corrected

Reviewer 2 Report
This review provides an overview of clinical trials of drug treatment for heart failure with preserved ejection fraction (HFPEF) and illustrates the limitations of classifying the pathophysiology by left ventricular ejection fraction (LVEF). As it includes topics from recent clinical trials and is likely to be of interest to readers, publication in this journal is worthy of consideration.
However, several points could be improved, which are listed below.
1) I agree with the author that LVEF is essential for a broad picture of the condition, but as indicated in the text, it is not suited to the assessment of essential cardiac function because of the many problems inherent in it. One essential comment on this review is that it should then be made clear what alternative measures are needed to describe cardiac function in the future. LVEF is important in distinguishing between HFrEF and non-HFrEF, given the obvious differences in the efficacy of heart failure drugs. As stated by the author, non-HFrEF patients are highly heterogeneous. They do not present a typical clinical picture, so, unsurprisingly, differentiation by LVEF does not offer any further promise. The stratification of non-HFrEF into treatment-responsive (similar to HFrEF) and non-responsive groups is useful for treatment prediction. What indicators are currently available for this? Does simply mentioning the problems with LVEF not solve the problem of erroneously assessing cardiac function?
2.SGLT2 inhibitors improve the prognosis of heart failure patients across the LVEF spectrum. A recently reported pooled analysis of EMPEROR-PRESERVED and DELIVER has shown that they reduce cardiovascular death or heart failure hospitalization even when LVEF is >60% (PMID: 36041474). Other than that, this paper has shown significant risk reduction with SGLT2 inhibitors even when stratified by age, sex, race, NYHA class, NT-proBNP level, KCCQ score, diabetes, BMI, and renal function, so it is unclear in what patient groups the benefit of SGLT2 inhibitors is stratified. Nevertheless, the fact that SGLT2 inhibitors are effective in a wide range of heart failure patients does not mean that LVEF classification is irrelevant: the distinction between HFrEF and non-HFrEF may still be useful in predicting treatment response, as the effectiveness of SGLT2 inhibitors is enhanced when combined with other heart failure therapies. Should the issue of LVEF and the broader effects of SGLT2 inhibitors be discussed in the same context?
3. line 94: It would be better to omit details of the CHS study, as it is easier to understand in the context of what has been said so far if the I-PRESERVED study is continued below.
4. line 34 of the abstract and line 443 of the text are based on which content in the text - is it consistent with line 193?
5. figure 1: this figure is misleading because SGLT2 inhibitors are also effective in HFrEF.
6. figure 2: The text in the figure is missing. It is not even clear what this diagram means in the first place. It is also unclear which part of the text it corresponds to.
7. line 323: it is not clear what this sentence means.
8. Is the sentence starting with "A renal benefit" necessary in context?
9. "Recent trial" is an unclear term: does ARNI also have a consistent effect regardless of LVEF?
10. There are many typographical errors and omissions in the manuscript throughout. This is extremely critical to ensure the quality of the paper. As peer review is not a proofreading service, we would like to avoid pointing out all the problems. While it is natural for the author to revise the paper again, there are too many points that need to be improved, so we would like to suggest that the paper should be revised by an external proofreading service.
Unnecessary double spaces (e.g. line 159)
Typographical errors (lines 160 and 380)
Unnecessary capitalization (line 217)
Use of decimal commas (line 345)
Unnecessary character insertions (line 374)
Unknown abbreviations (line 329)
Inclusion of British English (line 254)
Missing parentheses (line 411)
Author Response
this review provides an overview of clinical trials of drug treatment for heart failure with preserved ejection fraction (HFPEF) and illustrates the limitations of classifying the pathophysiology by left ventricular ejection fraction (LVEF). As it includes topics from recent clinical trials and is likely to be of interest to readers, publication in this journal is worthy of consideration.
Many thanks for your general appreciation, we tried to response to your comment accordingly
However, several points could be improved, which are listed below.
- I agree with the author that LVEF is essential for a broad picture of the condition, but as indicated in the text, it is not suited to the assessment of essential cardiac function because of the many problems inherent in it. One essential comment on this review is that it should then be made clear what alternative measures are needed to describe cardiac function in the future. LVEF is important in distinguishing between HFrEF and non-HFrEF, given the obvious differences in the efficacy of heart failure drugs. As stated by the author, non-HFrEF patients are highly heterogeneous. They do not present a typical clinical picture, so, unsurprisingly, differentiation by LVEF does not offer any further promise. The stratification of non-HFrEF into treatment-responsive (similar to HFrEF) and non-responsive groups is useful for treatment prediction. What indicators are currently available for this? Does simply mentioning the problems with LVEF not solve the problem of erroneously assessing cardiac function?
Many thanks for this comment that find us completely in line with your position. The problem of HFpEF classification or non HFrEF as you like to define, is the poor clinical and phenotype characterization. The sole EF categorization cannot distinguish among different patterns and additional clinical iconographic and laboratory features are necessary to recognize different phenotypes into HFpEF patients. In our paper we focused on LVEF considered as parameter of systolic function, but combined assessment of diastolic and detailed systolic function may help in risk stratification . We briefly reported this concept at page 15 adding two new references. Finally a refernce n 73 related to the special issue has been inserted
2.SGLT2 inhibitors improve the prognosis of heart failure patients across the LVEF spectrum. A recently reported pooled analysis of EMPEROR-PRESERVED and DELIVER has shown that they reduce cardiovascular death or heart failure hospitalization even when LVEF is >60% (PMID: 36041474). Other than that, this paper has shown significant risk reduction with SGLT2 inhibitors even when stratified by age, sex, race, NYHA class, NT-proBNP level, KCCQ score, diabetes, BMI, and renal function, so it is unclear in what patient groups the benefit of SGLT2 inhibitors is stratified. Nevertheless, the fact that SGLT2 inhibitors are effective in a wide range of heart failure patients does not mean that LVEF classification is irrelevant: the distinction between HFrEF and non-HFrEF may still be useful in predicting treatment response, as the effectiveness of SGLT2 inhibitors is enhanced when combined with other heart failure therapies. Should the issue of LVEF and the broader effects of SGLT2 inhibitors be discussed in the same context?
We reported the cited paper in the text (Ref N.77)and we agree that SGLT2 reffects is probably independent on different LVEF. However, the SGLT” beneficial effect are on top on traditional therapy Whereas the sole administration of these agents is not tested. Because of their ancillary effects SGLT2 may be useful even in early HF stage before the symptoms occurrence and in high risk patients.This is due to the positive vascular metabolic and renal effects but this is not the field of our paper that would discuss general Trials in HFpEF to demonstrate different enrollment criteria and phenotypes leading to contrasting result in interventional studies.
- line 94: It would be better to omit details of the CHS study, as it is easier to understand in the context of what has been said so far if the I-PRESERVED study is continued below. Patients details of CHS study have been deleted
- line 34 of the abstract and line 443 of the text are based on which content in the text - is it consistent with line 193?
- figure 1: this figure is misleading because SGLT2 inhibitors are also effective in HFrEF. The upper side of figure would describe then drugs commonly used in all HF patients and it is focused on gaps in EF. What the reviewer suggest is appropriately reported in Fiigure 2
- figure 2: The text in the figure is missing. It is not even clear what this diagram means in the first place. It is also unclear which part of the text it corresponds to. Changed inserting renal effects in both HFpEF and HFrEF
- line 323: it is not clear what this sentence means.
- Is the sentence starting with "A renal benefit" necessary in context?
- "Recent trial" is an unclear term: does ARNI also have a consistent effect regardless of LVEF?
- There are many typographical errors and omissions in the manuscript throughout. This is extremely critical to ensure the quality of the paper. As peer review is not a proofreading service, we would like to avoid pointing out all the problems. While it is natural for the author to revise the paper again, there are too many points that need to be improved, so we would like to suggest that the paper should be revised by an external proofreading service. thank, the paper is now corrected by an English native collaborator, therefore typos and grammar mistakes have been corrected

Round 2
Reviewer 2 Report
I reviewed the authors' replies and the revised paper. While I did not receive a proper response on the essential concerns regarding the manuscript, I could grasp what the authors claim.
Finally, minor errors in the manuscript to keep it presentable as a scientific paper should be corrected in order to maintain the appearance of a scientific paper. The first author is responsible for reporting the document correctly. Please recheck your text carefully, as not all the points raised by native English-speaking co-authors are correct. Once the authors are confident that all the problems I have previously pointed out have been resolved, it is considered a final version of the manuscript.
Author Response
2nd Round Reviewer 2
I reviewed the authors' replies and the revised paper. While I did not receive a proper response on the essential concerns regarding the manuscript, I could grasp what the authors claim.
We apologize our response is not completely satisfactory for you, I personally reviewed the previous answer to reviewer and the text and I found a point by point response. Thus to be honest, I don’t completely understand your second comments. I highlighted the specific response into the text in red colour and bold font so you can (Hopefully) appreciate our responses.
Specifically at page 10 lane 366 and page 11 lane 380 the reviewer could find our answer to previos concerns. I modified the Figure 1 including the 5 drugs universally endorsed for HF treatment irrespective of LVEF
Finally, minor errors in the manuscript to keep it presentable as a scientific paper should be corrected in order to maintain the appearance of a scientific paper. The first author is responsible for reporting the document correctly. Please recheck your text carefully, as not all the points raised by native English-speaking co-authors are correct. Once the authors are confident that all the problems I have previously pointed out have been resolved, it is considered a final version of the manuscript.
We rechecked the manuscript and typos oversights, All the grammar changed are placed in red font
